# Evaluating Causal Discovery Algorithms Without Ground Truth in Additive Noise Models

## Abstract

Evaluating the performance of causal discovery algorithms is a fundamental challenge in the field of causality. Most performance measures, such as Structural Hamming Distance (SHD) and Structural Intervention Distance (SID), rely on comparing the discovered graph with the ground truth, which is often unavailable in real-world applications. In this paper, we propose a novel evaluation measure called $\text{PIM}_{\text{score}}$, based on the Principle of Independent Mechanisms, for restricted Additive Noise Models (ANMs) and Linear Non-Gaussian Acyclic Models (LiNGAMs) to assess whether a given graph represents the true underlying structure based solely on observational data. The proposed measure aggregates the mutual information between the residuals obtained by regressing each variable on its parents in the given graph. We show that the true underlying graph achieves the lowest score in terms of this measure. In particular, we show that the mutual information among residuals is zero if and only if the given graph is the true one. Additionally, we introduce a method for leveraging this performance measure to rank the performance of a set of graphs based on how well they represent the ground truth. We evaluate the performance of our proposed measure using both synthetic and real data in discovering the true graph and compare its performance against two other baseline measures. Experimental results show that our proposed measure exhibits higher correlations with SHD and SID compared to existing approaches, making it a promising measure for evaluating recovered graphs when the true graph is unavailable.

## 1 Introduction

Identifying causal relationships is a key scientific objective in many disciplines (Pearl, 2009). Although Randomized Controlled Trials (RCTs) serve as the primary method to uncover these relationships, their implementation is not always practical due to ethical or legal reasons (Peters et al., 2017). Causal discovery aims to reveal the underlying mechanisms that govern observational data, distinguishing between cause-and-effect relationships and spurious correlations (Spirtes et al., 2000).

Causal discovery methods can be categorized according to their approach to learning causal graphs. Classical approaches include constraint-based methods, such as PC (Spirtes et al., 2000) and Fast Causal Inference (FCI) (Spirtes et al., 2000), score-based methods, such as Greedy Equivalence Search (GES) (Chickering, 2002), and more recent continuous-optimization approaches, such as NOTEARS (Zheng et al., 2018) and GOLEM (Ng et al., 2020). In general, without additional modeling assumptions, these methods can identify the causal structure only up to an equivalence class. Imposing additional functional-form assumptions on the causal model can enable unique identification of the true causal graph from purely observational data. Examples include Linear Non-Gaussian Acyclic Models (LiNGAMs) (Shimizu et al., 2006), which assume linear relationships with non-Gaussian exogenous noises, and Additive Noise Models (ANMs) (Hoyer et al., 2008), which allow for nonlinear functions with additive exogenous noises.

Regardless of the algorithm used, one of the main challenges in causal discovery is evaluating the performance of causal discovery algorithms. Because the true underlying causal graph is usually unknown in real-world applications, causal discovery is inherently an unsupervised learning problem. This is the case in many practical scenarios, for example, in hyperparameter optimization for tuning the hyperparameter configuration

of a causal discovery algorithm, or in model selection for choosing among several available causal discovery algorithms. Existing approaches for evaluation of causal discovery often rely on synthetic data or a limited number of benchmark datasets with known causal structures and using standard metrics such as Structural Hamming Distance (SHD) (Tsamardinos et al., 2006), Structural Intervention Distance (SID) (Peters & Bühlmann, 2015), and Adjustment Identification Distance (AID) (Henckel et al., 2024). However, in the more realistic setting (e.g., for hyperparameter optimization), such metrics cannot be applied directly as the ground truth graph is not available. This raises a fundamental question: how can we evaluate and compare the outputs of causal discovery algorithms when the true underlying graph is unavailable?

In this paper, we focus on two problems related to the evaluation of causal discovery algorithms:

- **Causal Graph Verification ($G_{\mathbf{ver}}$) problem:** Given a set of candidate causal graphs $\mathcal{G}$ (assumed to contain the true graph) and observational data $\mathcal{D}$, determine which graph $G$ represents the true underlying causal structure.

- **Causal Graph Ranking ($G_{\mathbf{rank}}$) problem:** Given the observational data $\mathcal{D}$ and a set of candidate causal graphs $\mathcal{G}$ produced by different causal discovery methods (or hyperparameter configurations), the goal is to rank these graphs using only $\mathcal{D}$ so that the ranking reflects their true closeness to the unknown ground-truth, as measured by a distance such as SHD or SID.

In order to assess whether a graph is the true underlying causal structure (in $G_{\mathrm{ver}}$ problem), we rely on the Principle of Independent Mechanisms (PIM) (Peters et al., 2017), as a fundamental principle in structural causal models, stating that the process generating the system's variables involves independent modules that operate without impacting one another. PIM implies that the conditional distribution of the effect given the cause should not change by altering the cause distribution. Deviations from PIM can be used as a measure for quality of a graph.

In this paper, we propose a novel score utilizing PIM for LiNGAMs (linear case) and ANMs (nonlinear case) to address both $G_{\mathrm{ver}}$ and $G_{\mathrm{rank}}$ problems from merely observational data. In particular, we make the following contributions:

- Assuming that the data follows the LiNGAM or ANM assumptions (with nonlinear structural functions), we propose a score, called the $\mathrm{PIM}_{\mathrm{score}}$, which evaluates a graph $G$ by regressing each variable on its parents in the graph and measuring the independence among the resulting residuals through estimating mutual information. Specifically, the score reflects how independent the residuals of different variables are from one another. For the $G_{\mathrm{ver}}$ problem, under the causal minimality assumption, we prove for both LiNGAM and ANM that only the true graph attains the lowest score (Theorem 1).

- We extend the $\mathrm{PIM}_{\mathrm{score}}$ to $G_{\mathrm{rank}}$ problem by introducing two aggregation schemes, based on the average and the maximum of the mutual information measures associated with a given graph. In the LiNGAM case, we analyze how errors in a causal graph, such as missing or extra edges, induce dependence among residuals, which provides support for the average aggregation scheme of mutual information measures.

- Experimental results on synthetic data generated based on linear (LiNGAM) and nonlinear (ANM) models and real data show that, for $G_{\mathrm{ver}}$, the proposed score outperforms existing methods on graphs of moderate to large size. Moreover, for $G_{\mathrm{rank}}$, it exhibits a stronger correlation with SHD and SID, making it a more effective proxy for ranking a set of candidate graphs based on their SHDs or SIDs.

## 2   Related Work

In the following, we review related work and available measures for evaluating the quality of causal discovery based on the type of available information and its respective applications.

**Evaluation with Ground Truth:**

When ground truth is available, evaluation of causal discovery can be done using metrics used in binary classification. The causal structures can be represented using an adjacency matrix, where nonzero entries indicate the presence of directed edges. In this case, standard classification metrics, such as accuracy, true positive rate, false positive rate, precision, and recall, can be employed to evaluate the performance of the discovered graph compared to the original graph (Hasan et al., 2023).

SHD and SID are other popular metrics used for this purpose. SHD (Tsamardinos et al., 2006), counts edge additions, deletions, and reversals between the estimated graph and the truth, providing a single-value summary metric. SID (Peters & Bühlmann, 2015) is another metric providing information that is more in line with downstream tasks (e.g., causal effect identification), showing the number of variable pairs for which the estimated graph suggests a set of interventional distributions different from those implied by the true graph using a purely graphical criterion. Adjustment Identification Distance (AID) (Henckel et al., 2024) quantifies causal dissimilarity by counting how often an adjustment set inferred from an estimated graph fails to identify the true causal effect. SID can be interpreted as a specific form of AID. As SHD and SID provide different information about the differences between the two graphs, we will use both as a reference in our work.

**Evaluation without Ground Truth:**

In many real-world applications, the true underlying causal graph is not available. Therefore, moving towards real-world applicability, having access to a measure of the quality of the graph is critical. Faller et al. (2024) suggest two scores for the $G_{\mathrm{rank}}$ problem, Interventional Incompatibility, and Graphical Incompatibility (GI). In this approach for a given algorithm and data, subsets of variables are randomly selected, then the algorithm is applied on the whole data (joint graph) and selected subsets (marginal graphs). As the names suggest, the method looks for incompatibilities in the joint graph and marginal graphs. Interventional incompatibility is based on assumptions that rely on linearity of the model, and is closer to SID. GI is SHD-like because it counts graph-edit discrepancies, but it compares joint-vs-marginal learned graphs rather than comparing to the ground truth.

It is worth noting that these measures address the $G_{\mathrm{rank}}$ problem since they do not require access to the ground-truth graph and can serve as proxies for SHD. However, they rely on repeatedly running a causal discovery algorithm on multiple subsets of the observed variables. As a result, they cannot be directly applied to the $G_{\mathrm{ver}}$ problem.

**Algorithm Configuration in Causal Discovery:**

Causal discovery algorithms have several hyperparameters that can affect the result and need to be fine-tuned without access to the ground truth. Since algorithm configuration requires an evaluation step, several ground-truth-free measures have been proposed, in which the goal is to select a hyperparameter setting using an unsupervised evaluation score (see e.g., (Biza et al., 2024)). Although these measures have been used for algorithm configuration, they lack theoretical guarantees and, without modification, are not applicable to $G_{\mathrm{ver}}$.

Stability Approach to Regularization Selection (StARS) (Liu et al., 2010) was originally introduced to tune the hyperparameters of the graphical lasso, aiming to select the most stable network with respect to small perturbations of the input data. More recently, a modification of StARS (Biza et al., 2024) has been proposed to tune hyperparameters in causal discovery algorithms. For a given configuration, the probability of the presence of an edge is estimated by learning multiple graphs from resamples of the data without replacement. This probability is used to define a measure of instability of each edge and the total network instability for a configuration as the average instability over all possible edges. Finally, a configuration with instability below a certain threshold is selected. StARS accounts for outliers or small perturbations in the data but does not consider the quality of the graph and how it fits to the data; therefore, a configuration that consistently identifies a graph with higher SHD output (or any other measures) is preferred over a configuration that identifies a better fit but is not necessarily consistent.

Out-of-sample Causal Tuning (OCT) (Biza et al., 2024) is another approach proposed for hyperparameter tuning of causal discovery algorithms. For the evaluation of the quality of the hyperparameter configuration used to derive the causal graph, OCT treats each variable in the graph as a target of a different prediction task. The Markov boundary of each node $X$, obtained from the discovered graph, is used to train a predictive model on the training set and predict $X$ on the test set. The mutual information between each variable and its prediction is then computed for the test set, and the average of mutual information across all variables is used as a performance measure for the configuration. To avoid overly dense graphs, a sparsity penalty based on a permutation test is applied to select a configuration after all configurations are evaluated. Since the Markov boundary of a variable is the same as in all graphs in the Markov Equivalence Class (MEC), OCT gives the same score to all Markov equivalent graphs. However, OCT might give the same score for graphs that are not Markov equivalent (see an example in the Figure B.1).

Both OCT and StARS are designed for selecting a hyperparameter configuration in the context of hyperparameter optimization and not necessarily to address the $G_{\mathrm{ver}}$ and $G_{\mathrm{rank}}$ problems. The first part of OCT, i.e., the predictive performance of a graph, can be used as a measure for both $G_{\mathrm{ver}}$ and $G_{\mathrm{rank}}$. However, StARS cannot evaluate the quality of a single graph, as it measures the instability of edges among a set of graphs linked to a configuration and not a single graph.

## 3 Notations, Preliminaries, and Problem Definition

In this section, we introduce the notations and definitions, review ANM and LiNGAM, and state the problem definition. In particular, Section 3.1 fixes our notation, definitions, and symbols; Section 3.2 introduces the models in this study; and Section 3.3 gives a precise problem definition.

### 3.1 Notations

We denote random variables by capital letters and values of a random variable with the respective lowercase letter. Vectors and matrices are represented in boldface, and all vectors are assumed to be column vectors. The entry at position $(i, j)$ of a matrix $\mathbf{A}$ is denoted as $[\mathbf{A}]_{i,j}$.

A directed graph is defined as a pair $G = (V, E)$, where $V = \{X_1, \ldots, X_p\}$ represents the set of nodes and $E \subseteq \{(i, j) \mid X_i, X_j \in V, i \neq j\}$ is the set of edges.

A directed path from node $X_i$ to node $X_j$ in a graph $G$ is defined as a sequence of nodes $\pi = (i_1 = i, i_2, \ldots, i_{k+1} = j)$ such that $(i_p, i_{p+1}) \in E$ for $p \in \{1, \ldots, k\}$. A cycle in $G$ is a path that starts and ends at the same node $X_i$. A Directed Acyclic Graph (DAG) is a directed graph that contains no cycles.

We say that $X_i$ is a parent of $X_j$ if there is an edge $(i, j) \in E$, while $X_j$ is a child of $X_i$. If there exists a path from $X_i$ to $X_j$ in $G$, we refer to $X_i$ as an ancestor of $X_j$ and $X_j$ as a descendant of $X_i$. The sets of parents, children, ancestors, and descendants of a given node $X_i$ in graph $G$ are denoted by $pa_G(X_i)$, $ch_G(X_i)$, $an_G(X_i)$, and $de_G(X_i)$, respectively. We append a superscript $*$ to indicate reference to the true causal graph (ground truth).

A Structural Causal Model (SCM) (Pearl, 2009) is a set of equations that describe the data-generating process of each variable $X_j$ given its parents $pa_G(X_j)$. In particular,

$$X_j := f_j(pa_G(X_j), N_j), \qquad 1 \leq j \leq p, \tag{1}$$

where $f_j$ is the functional mechanism for generating $X_j$ and $N_j$ is the corresponding exogenous noise of $X_j$. We assume that the model is causally sufficient, that is, there are no latent confounders. Moreover, the exogenous noises $N_1, \ldots, N_p$ are jointly independent.

### 3.2 Models

One of the most common modeling assumptions in causal discovery is ANM (Peters et al., 2017), which posits that each variable is a function of its parents plus an exogenous noise term independent of the parents. In

particular, in ANMs, each variable is modeled as:

$$X_j = f_j(\mathrm{Pa}(X_j)) + N_j, \qquad 1 \leq j \leq p, \tag{2}$$

where $f_j$ is a (potentially nonlinear) function, and $N_j$ is an additive exogenous noise term; exogenous noises are assumed to be jointly independent across variables (Hoyer et al., 2008; Peters et al., 2014). In our theoretical analysis, we consider the class of restricted ANMs (Peters et al., 2014), in which the exogenous noises are assumed to have non-vanishing densities and the functions $f_j$ are assumed to be continuous and three times continuously differentiable.

LiNGAM is a special case of ANM where $f_j$ is linear for all $j$, and exogenous noises are non-Gaussian. In the linear case, the structural equations can be written in vector form as:

$$\mathbf{X} = \boldsymbol{B}\mathbf{X} + \mathbf{N}, \tag{3}$$

where $\mathbf{X}$ and $\mathbf{N}$ are the vectors of random variables $X_j$'s and $N_j$'s, respectively. Moreover, $[\boldsymbol{B}]_{j,i} = 0$ if $(i, j) \notin E$, and it shows the direct causal effect of $X_i$ on $X_j$.

In the linear models, one persistent challenge is identifiability; if the noise variables are all Gaussian, only the Markov equivalence class can be recovered from observational data. If the exogenous noises are non-Gaussian (at most one Gaussian noise term), the full causal DAG can be identified (Shimizu et al., 2006). Hoyer et al. (2008) showed that nonlinearities act similarly to non-Gaussianity and provided an identifiability result for ANMs.

### 3.3 Causal Model Verification Problems

In the following, we define two problems of Causal Graph Verification $G_{\mathrm{ver}}$ and Causal Graph Ranking $G_{\mathrm{rank}}$.

Let $\mathcal{H}$ be the set of all possible DAGs on $V$. Let $\mathcal{D} = \{x^{(1)}, x^{(2)}, \dots, x^{(n)}\}$ denote observational data drawn i.i.d. from an unknown distribution $P_V$ over $V = \{X_1, \dots, X_p\}$, where $P_V$ is Markov and faithful with respect to a ground-truth DAG $G^* \in \mathcal{H}$.

**Problem 1.** *Let $\mathcal{G} \subseteq \mathcal{H}$ be a finite set of candidate DAGs, assumed to contain the ground-truth graph, $G^* \in \mathcal{G}$. Given only the observational data $\mathcal{D}$, the goal is to identify which graph in $\mathcal{G}$ is the ground-truth graph.*

**Problem 2.** *Let $\mathcal{G} \subseteq \mathcal{H}$ be a finite set of candidate DAGs obtained by some causal discovery algorithms applied to $\mathcal{D}$. Let $F : \mathcal{H} \times \mathcal{H} \to \mathbb{R}_{\geq 0}$ be a score function, representing a distance to ground truth (such as SHD or SID). **Having access only to** $\mathcal{D}$, the goal is to output a ranking on $\mathcal{G}$ that is consistent with the unknown scores $\{F(G, G^*)\}_{G \in \mathcal{G}}$.*

## 4 Method

In this section, we introduce our proposed scoring framework to address $G_{\mathrm{ver}}$ and $G_{\mathrm{rank}}$ problems. Our approach is fundamentally motivated by the PIM (Peters et al., 2017), which holds that the causal generative process of the variables in the SCM is composed of autonomous modules that do not inform or influence each other. Accordingly, we propose a scoring method based on the independence of estimated exogenous noise terms. While the connection of PIM to the assumption of independent noise terms is less obvious (for details, refer to (Peters et al., 2017) Chapter 2.1), it motivates scoring via exogenous noise estimate independence. In particular, we first establish a theoretical guarantee: under ANM identifiable model classes (such as LiNGAMs and Restricted ANMs), any incorrect candidate graph induces statistical dependence among the estimates of its exogenous noises, and then introduce our score.

### 4.1 Theoretical Foundations

We first formalize the theoretical foundations of our proposed score. Assuming causal sufficiency (i.e., no unobserved confounders) and based on the PIM, for the true graph, we know that the induced exogenous

noises are mutually independent. However, relying on this property to solve $G_{\text{ver}}$ requires proving the converse: no other graph exhibits this property, i.e., at least one pair of exogenous noises becomes dependent under any incorrect graph. Proving this via an explicit algebraic derivation of the residuals is feasible for LiNGAMs but becomes complex for Additive Noise Models (ANMs) with nonlinear mechanisms. For the proof, we consider a general framework that applies to any ANM following Equation 2 that has identifiability results and establish that with an appropriate regression method that adheres to the rules of the framework, only the ground truth will show independence between the residuals. We then instantiate this generic lemma to provide concrete guarantees for both LiNGAMs and Restricted ANMs.

**Definition 1.** *Causal Minimality: A distribution satisfies causal minimality with respect to $G$ if it is Markovian with respect to $G$, but not to any proper subgraph of $G$.*

Let $\mathcal{M}$ be an additive-noise model class that is identifiable from the observational distribution among causally minimal representations (e.g., LiNGAM, restricted ANMs). For the population distribution $\mathbb{P}(\mathbf{X})$ over a set of variables $\mathbf{X}$, we assume:

1. **Assumption 1 (regarding data-generating process):** $\mathbb{P}(\mathbf{X})$ admits a representation in $\mathcal{M}$ over a true causal minimal DAG $G^*$, where $X_j = f_j(\text{Pa}_{G^*}(X_j)) + N_j$. The noise variables $N_j$ are jointly independent. Without loss of generality, we absorb any constant mean into $f_j$ such that $\mathbb{E}[N_j \mid \text{Pa}_{G^*}(X_j)] = 0$.

2. **Assumption 2 (regarding induced representation):** For any candidate DAG $G$, let $m_j(\text{Pa}_G(X_j)) := \mathbb{E}[X_j \mid \text{Pa}_G(X_j)]$ denote the population regression functions, and let $R_j := X_j - m_j(\text{Pa}_G(X_j))$ be the corresponding residuals. We assume the induced representation from regression satisfies the structural constraints of $\mathcal{M}$ (e.g., functional form, density requirements) such that, provided the residuals are jointly independent, and the representation forms a causal minimal member of $\mathcal{M}$.

Thus, Assumption 2 does not introduce a new identifiability condition beyond those required by $\mathcal{M}$; rather, it restricts the residual-based evaluation to candidate decompositions that fall within the identifiable regime of that model class.

**Theorem 1.** *Under Assumptions 1 and 2, for any candidate DAG $G$, the population residuals $R_1, \ldots, R_p$ are jointly independent if and only if $G = G^*$.*

*Proof. If $G = G^*$, then the residuals are jointly independent:* Assume $G = G^*$. By Assumption 1, the variables are generated as $X_j = f_j(\text{Pa}_{G^*}(X_j)) + N_j$. Because the noise terms $N_j$ satisfy $\mathbb{E}[N_j \mid \text{Pa}_{G^*}(X_j)] = 0$, we can expand the population conditional expectation as:

$$\begin{aligned}
\mathbb{E}[X_j \mid \text{Pa}_{G^*}(X_j)] &= \mathbb{E}[f_j(\text{Pa}_{G^*}(X_j)) + N_j \mid \text{Pa}_{G^*}(X_j)] \\
&= f_j(\text{Pa}_{G^*}(X_j)) + \mathbb{E}[N_j \mid \text{Pa}_{G^*}(X_j)] \\
&= f_j(\text{Pa}_{G^*}(X_j)).
\end{aligned}$$

The population residuals for $G^*$ are therefore:

$$R_j = X_j - \mathbb{E}[X_j \mid \text{Pa}_{G^*}(X_j)] = N_j. \tag{4}$$

Since the true noise terms $N_j$ are jointly independent, the residuals $R_j$ are jointly independent.

*If the residuals are jointly independent, then $G = G^*$:* Assume the population residuals $R_1, \ldots, R_p$ for a candidate graph $G$ are jointly independent. By definition of the residuals, we can write the following structural equations:

$$X_j = m_j(\text{Pa}_G(X_j)) + R_j, \quad j = 1, \ldots, p. \tag{5}$$

Hence, $\mathbb{P}(\mathbf{X})$ admits an additive-noise representation over $G$ with structural functions $m_j$ and jointly independent noise variables $R_j$. By Assumption 2, this induced representation is a causally minimal member of $\mathcal{M}$. Therefore, $\mathbb{P}(\mathbf{X})$ admits two causally minimal representations in $\mathcal{M}$, one with graph $G^*$ and one with graph $G$. By identifiability in $\mathcal{M}$, these DAGs must coincide. Hence $G = G^*$. $\qquad\square$

Theorem 1 isolates the core theoretical burden of our work via residual independence. We instantiate this principle for two prominent identifiable causal frameworks below.

**Corollary 2** (Uniqueness in LiNGAM). *Let $\mathcal{M}$ be the class of LiNGAMs (Shimizu et al., 2006). If the data-generating process is a causally minimal LiNGAM over $G^*$, and for a candidate graph $G$, the induced representation is also a causally minimal LiNGAM, then the residuals $R_1, \ldots, R_p$ are jointly independent if and only if $G = G^*$.*

**Corollary 3** (Uniqueness in Restricted ANMs). *Let $\mathcal{M}$ be the class of Restricted ANMs (Peters et al., 2014). If the data-generating process is a causally minimal Restricted ANM over $G^*$, and for a candidate graph $G$, the induced representation is a causally minimal restricted additive-noise model, then the residuals $R_1, \ldots, R_p$ are jointly independent if and only if $G = G^*$.*

By regressing each variable on its parents, if the given graph embodies the true underlying causal structure and an appropriate regression method is used (i.e., linear regression for linear data), it is expected that the exogenous noise of that variable will be the only term in the residual, and all the residuals must be independent due to PIM.

**Remark 1.** *Our algebraic derivation for LiNGAM (details in Appendix A) shows that, similar to SHD, each error in the estimated graph introduces at least one dependency among the residuals. However, the strength of this dependency, as measured by mutual information, has not been characterized analytically. Nevertheless, we expect that in larger and denser graphs, where the number of pairwise comparisons and dependencies is higher, average aggregation over pairwise mutual information mitigates finite-sample variability and shows a high correlation with SHD.*

Next, we illustrate this point with an example, in which we show, algebraically, that an error in the structure creates at least one dependency among the residuals of regression.

**Example 1.** *Each structural error can be viewed as the addition or removal of a parent for a given node (for details, see the Appendix A). The case of having a non-ancestor missing parent: In a graph $G$ with ground truth of $G^*$, without loss of generality, consider variable $Y$ with one or more missing parents, consider a missing parent $X_j$, $X_j \in Pa_{G^*}(Y)$ and $X_j \notin Pa_G(Y)$, we denote the missing parents as $MPa_G(Y)$. In this case, we assume $X_j$ has no descendant in the parents, i.e. $De_{G^*}(X_j) \cap (Pa_{G^*}(Y) \backslash X_j) = \varnothing$. Let $\hat{Y}$ denote the fitted value obtained by regressing $Y$ on its parents in $Pa_G(Y)$, and $\beta_i$ are the respective coefficients from the regression. We have:*

$$Y - \hat{Y} = \sum_{X_i \in Pa_{G^*}(Y)} \beta_i^* X_i + N_Y - \sum_{X_i \in Pa_G(Y)} \beta_i X_i \tag{6}$$

$$= \sum_{X_i \in Pa_G(Y)} (\beta_i^* - \beta_i) X_i + N_Y + \tag{7}$$

$$\sum_{X_i \in \{MPa_G(Y) \backslash \{X_j\}\}} \beta_i^* X_i + \beta_j^* X_j$$

*As $X_j$ does not have any descendant in other parents of $Y$, $N_j$ is only present in $X_j$ and it will remain in residuals of regressing $Y$ on its parents; therefore, based on Darmois-Skitovic theorem (Darmois, 1953) by checking dependency of the residual of the regression with $X_j$ we have: Residual $\not\perp X_j$*

### 4.2 PIM$_{\text{score}}$

We design a score (given in Algorithm 1) for solving the $G_{\text{ver}}$ and $G_{\text{rank}}$ problems. In particular, the algorithm regresses each variable on its parents (lines 2-3), computes the residual of the regression (line 4), and identifies dependencies between pairs of residuals using mutual information (lines 6-7). The values of mutual information are stored in a matrix $\mathbf{M}$. We consider both maximum and average as aggregation functions over $\mathbf{M}$ as proxies for counting the number of missing/extra parents in the graph. We expect that the average summarizes dependencies across the whole graph, whereas the maximum captures the strongest residual dependence. Therefore, the maximum is a more conservative aggregation.

---

**Algorithm 1**

---

**Require:** Graph $G$, Dataset $D$. Let $N$ be the set of all variables in $G$.

1: **for all** variables $X$ in $N$ **do**
2:      Regress $X$ on $pa_G(X)$ using $D$ to fit a function $f$.
3:      $\hat{X} \leftarrow f(pa_G(X))$
4:      Compute residuals $R_X \leftarrow X - \hat{X}$
5: **end for**
6: **for all** ordered pairs $(V, W)$ such that $V, W \in N$ and $V \neq W$ **do**
7:      Compute $\mathbf{M}[V, W] \leftarrow \log(I(R_V; R_W))$                $\triangleright$ $I$: Mutual Information
8: **end for**
9: $\text{PIM}_{\text{score}} \leftarrow \text{aggregate}(\mathbf{M})$                        $\triangleright$ Maximum or Average
10: **return** $\text{PIM}_{\text{score}}$

---

The choice of regression method is data-driven. For linear data, the method is limited to linear regression. For nonlinear data, it should be able to express the functions that generate the data and adhere to conditions proposed by (Peters et al., 2014) (In some cases a reverse nonlinear model exists for a linear model, which is why LiNGAM $\not\subset$ Restricted ANMs). A practical challenge for Algorithm 1 concerns Assumption 2 in the theorem, namely causal minimality. Graphs that violate causal minimality must, in principle, be detected by the regression step. In linear models, this can often be handled by identifying zero regression coefficients. In nonlinear settings, however, detecting such violations is less straightforward, since redundant parent variables may not be easily identifiable from the fitted regression function. We therefore view the robust treatment of non-minimal graphs in the nonlinear case as a research direction for future work.

## 5 Experiments

In this section, we aim to answer the following two questions. First, we want to evaluate whether our score can identify the ground truth, i.e., $G_{\text{ver}}$ problem, and how it performs against other methods. Second, we want to evaluate the performance of our proposed method in ranking graphs generated from causal discovery algorithms, and also compare its performance with other methods. The code for experiments is available at the following link. [1]

Our experiments are focused on DirectLiNGAM (Shimizu et al., 2011) and NOTEARS (Zheng et al., 2018) for linear and NOTEARS-MLP (Zheng et al., 2020) for nonlinear settings. We use the implementation from the causal-learn library (Zheng et al., 2024) for DirectLiNGAM, and gCastle (Zhang et al., 2021) for NOTEARS and NOTEARS-MLP. For regression in $\text{PIM}_{\text{score}}$, we use linear regression in scikit-learn (Pedregosa et al., 2011) for linear cases and a neural network with one hidden layer (100 neurons) and ReLU activation for nonlinear cases. While a choice of tanh activation function could result in better approximation of functions, ReLU was chosen to show that the score can work with any universal approximator and does not need to be hand-tailored to the data.

In the following sections, we will introduce performance measures, present baselines, describe the experimental setup, and discuss the results.

### Performance Measures

We compare our proposed measure, $\text{PIM}_{\text{score}}$, against SHD, a widely accepted measure in the field, as well as SID, as performance measures and quality definitions. Since SHD and SID assess the quality of the causal graph with ground truth, our goal is to evaluate how effectively our method provides insights into these two metrics without access to ground truth. SHD between two graphs represents the total number of edge insertions, deletions, and reversals required to transform one into the other. SID quantifies the number of pairwise interventional distributions incorrectly inferred in the estimated graph compared to the reference graph (ground truth).

---

[1]https://anonymous.4open.science/r/PIMScore/

**Baselines**

As baselines, we consider measures that do not require ground truth to evaluate the quality of the causal graph.

- **Graphical Incompatibility** (Faller et al., 2024): We considered the GI score as it is applicable for both linear and nonlinear settings. As GI relies on causal discovery in subsets of data, we have used DirectLiNGAM (Shimizu et al., 2011) and NOTEARS-MLP (Zheng et al., 2020) for linear and nonlinear data causal discovery algorithms, respectively. For GI, we use the original implementation from the paper.

- **Out-of-sample Causal Tuning (OCT)** (Biza et al., 2024): OCT is proposed to evaluate the quality of hyperparameter configurations of causal discovery algorithms by measuring the predictive performance of the discovered graph. Once all possible configurations are evaluated, a sparsity penalty is applied to select the configuration with the sparsest graph comparable to the best-performing one. We considered the first step of OCT to quantify the graph quality and as a baseline. The sparsity penalty only allows selecting a graph among a set of graphs, and it does not provide a consistent ranking option for all graphs.

**Datasets**

We used both simulated and real datasets for our experiments.

**Simulated data:**

Following the common practice in causality research, we simulated datasets for our experiments. Unless stated otherwise, the synthetic data setup is as follows for all experiments. All DAGs are generated using the Erdős–Rényi random graph model (Erdős & Rényi, 1959). Functions are either linear or, in the case of nonlinear, the tanh function was used. Structural coefficients are sampled independently from a uniform distribution over $[-2, -0.5] \cup [0.5, 2]$ for linear models and between $[-1, -0.1] \cup [0.1, 1]$ for nonlinear. These ranges follow common synthetic causal discovery benchmarks and avoid near-zero edge weights. Exogenous noises are generated independently from a uniform distribution with zero mean and unit variance. Each dataset contains 10000 i.i.d. observations.

**Real data:**

We also evaluated our method on the Sachs dataset (Sachs et al., 2005), a standard real-world benchmark with known ground truth for causal discovery algorithms. This dataset contains 7466 single-cell measurements of 11 proteins with well-studied signaling pathways.

## 5.1 Causal Graph Verification

To assess the ability of our proposed measure and baseline scores to identify the ground truth, in both LiNGAM and ANM settings, we initially generate a DAG serving as the ground truth. Based on this DAG, we generate a dataset and 100 candidate DAGs. We compare the performance of the measures by comparing these 100 candidate DAGs with the original DAG. For each combination of graph size and edge density, 100 ground-truth DAGs with the same size and density are generated. To do so, we randomly modified the ground-truth DAGs to produce graphs with a 10% difference in the number of edges relative to the ground truth. This is a more realistic scenario, considering that the different candidate DAGs can be outputs of causal discovery algorithms. We used a causal discovery algorithm matching the data-generation process to calculate GI (i.e., NOTEARS and NOTEARS-MLP). We measure the performance for the $G_{\mathrm{ver}}$ problem as the fraction of cases in which the ground truth is ranked in the top-$N$ among all graphs. The Wilson score interval (Wilson, 1927) is used to calculate the 95% confidence intervals to assess the performance of identifying the ground truth.

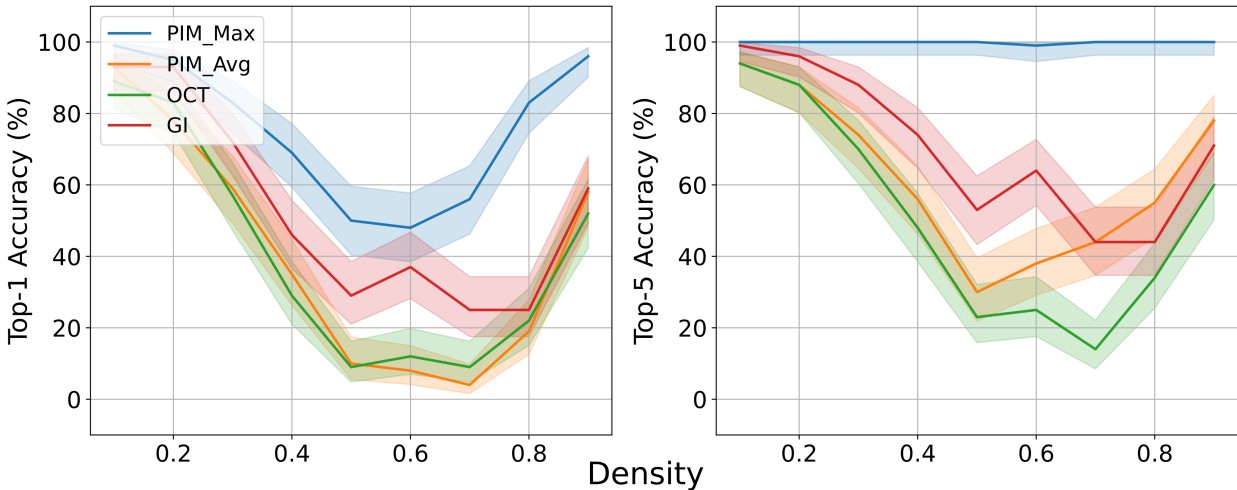

Figure 1: Top-1 (left) and Top-5 (right) accuracy of identifying the ground truth among similar graphs in LiNGAM) setting, for graphs with size 5.

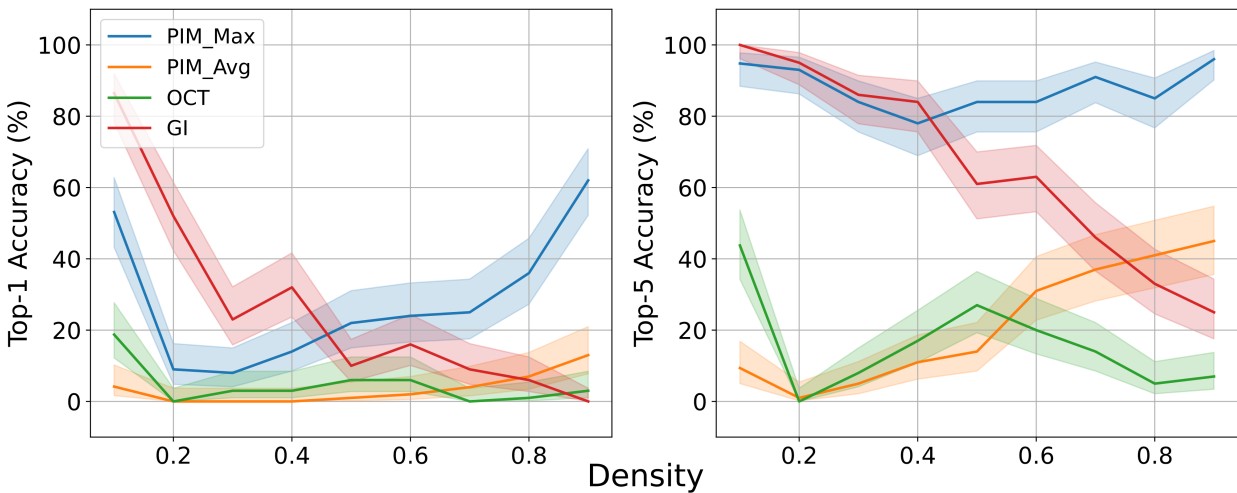

Figure 2: Top-1 (left) and Top-5 (right) accuracy of identifying the ground truth among similar graphs in LiNGAM setting, for graphs with size 10.

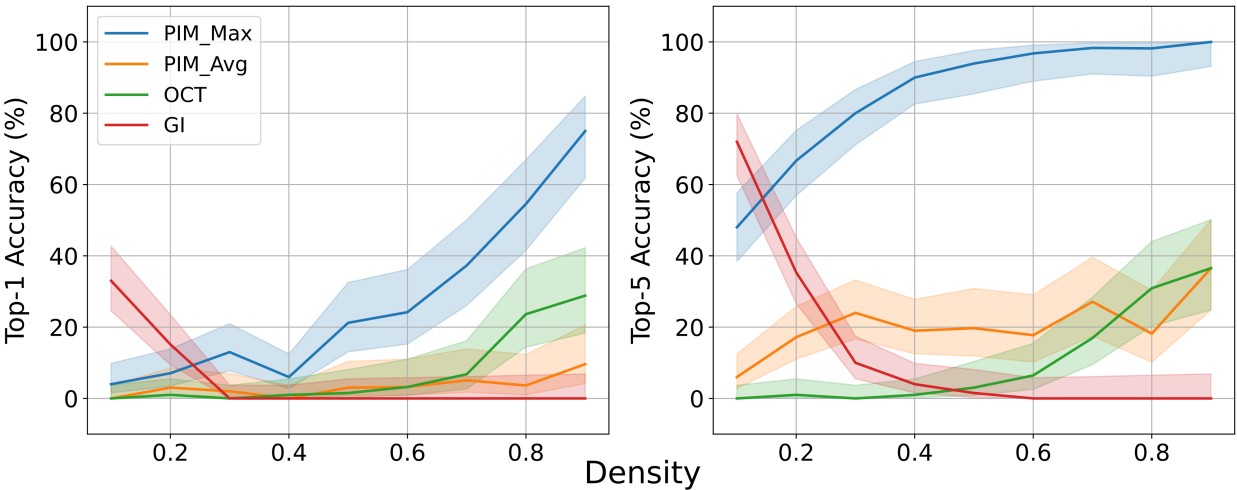

Figure 3: Top-1 (left) and Top-5 (right) accuracy of identifying the ground truth among similar graphs in ANM (nonlinear) setting, for graphs with size 5.

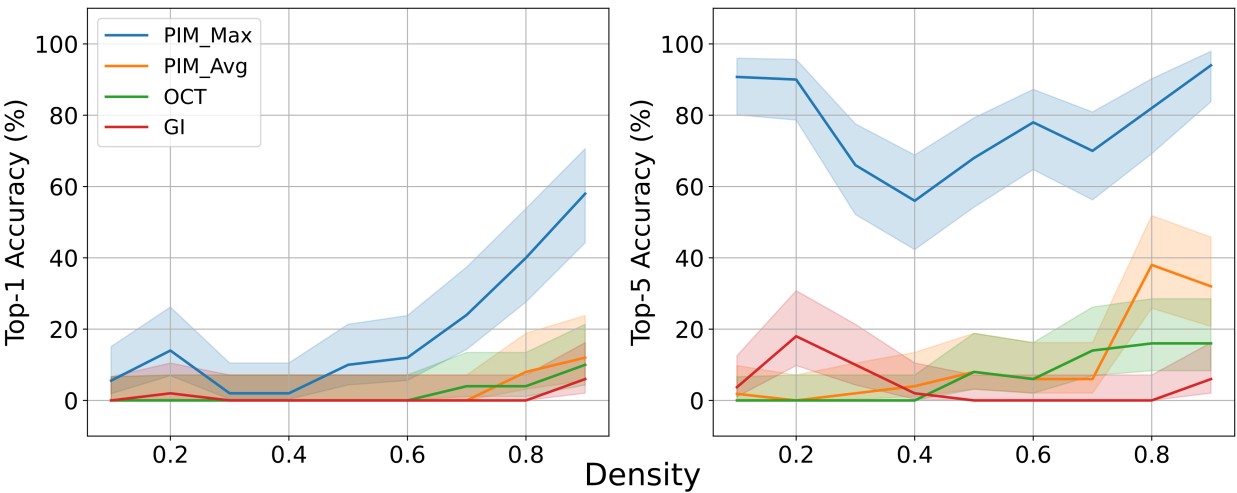

Figure 4: Top-1 (left) and Top-5 (right) accuracy of identifying the ground truth among similar graphs in ANM (nonlinear setting), for graphs with size 10.

Figure 1 and 2 depict the Top-1 (left) and Top-5 (right) performance of our method $PIM_{score}$ implemented with maximum and average aggregation functions (denoted by $PIM_{Max}$ and $PIM_{Avg}$, respectively), GI, and OCT in $G_{ver}$ problem, with graph sizes of 5 (Figure 1) and 10 (Figure 2) with densities ranging from 10% to 90% in LiNGAM setting. It can be seen that OCT has the lowest performance among all measures, competing with $PIM_{Avg}$ in some cases. While GI has better performance at lower densities, there is a performance drop at moderate and high densities. $PIM_{Max}$ has the best performance in all experiments with densities higher than 40% and in some cases in lower densities as well. While $PIM_{Avg}$ does not perform as well as $PIM_{Max}$, it maintains its performance with a rise in higher-density graphs.

Figure 3 and 4 show Top-1 (left) and Top-5 (right) accuracies in the nonlinear setting for graph sizes of 5 and 10. It can be seen that GI has zero accuracy in some settings. As an abnormal behavior, we investigated further and noticed these are cases where the causal discovery algorithm has failed and returned empty graphs.

Lower performance of the $PIM_{score}$ in low-density graphs can be understood by revisiting the second assumption of Theorem 1. In particular, we assume that the regression method is able to identify non-minimal decompositions. However, this assumption is not explicitly incorporated into the algorithm, mainly because of the additional challenges posed by nonlinear regression and by the difficulty of combining two different sources of information, namely regression weights and mutual information. At lower graph densities, candidate graphs have more room to include edges between variables that are in fact independent. If the method does not explicitly account for the regression weights, such unnecessary edges may not be properly detected, which can lead to errors.

While both max and average aggregation use the same data (pairwise mutual information between residuals), the better performance of max suggests that the information required for this problem is contained in the right tail, and averaging it dilutes it with noise. Another point that stands out in comparison between results in the linear and nonlinear settings is the lower performance of $PIM_{score}$ in the nonlinear setting compared to linear models. The performance of $PIM_{score}$ highly relies on the choice of regression method to approximate the data-generating function. Our choice for the nonlinear setting was a shallow network with the ReLU activation function, which might not be the best choice for approximating the tanh function used in our data generation process.

To conclude on our first research question, our proposed scores can be effective for verifying whether a graph represents the ground truth, and they outperform other methods on moderate to dense graphs.

## 5.2   Causal Discovery Ranking

To evaluate the methods for solving the $G_{rank}$ problem, we limit the space of DAGs, from all possible DAGs to only the DAGs that are more probable to be of interest. The target of this question is to assess the effectiveness of the measures for relevant applications, such as evaluating the output of causal discovery algorithms. We explore how each score is correlated to SHD and SID.

In these experiments, we first generate a DAG as the ground truth and generate a dataset based on this DAG, run causal discovery (DirectLiNGAM and NOTEARS-MLP) on the generated dataset, and evaluate the quality of the discovered graph against ground truth graph using the evaluation scores ($PIM_{score}$, OCT, GI). For this experiment, 100 and 50 ground-truth DAGs of the same size and density were generated for linear and nonlinear scenarios, respectively. Pearson correlations are computed between each score and SHD and SID for all generated DAGs, and confidence intervals are computed via bootstrapping (with replacement) 1000 times. We investigate graph sizes of 10, 20, and 100 in the linear setting, and sizes of 10 and 20 in the nonlinear setting due to computational costs. Graph size 5 was not used due to the high performance of causal discovery algorithms in identifying ground truth.

To assess whether the difference in correlation coefficients between our method and the baselines is statistically significant, we computed, over bootstrap samples, the difference between the correlation coefficient of our method and that of each baseline (OCT and GI). The resulting distribution of differences produces a 95% confidence interval. If this interval excludes zero, we can conclude that our proposed score exhibits a statistically significant improvement in alignment with SHD and SID compared to a baseline.

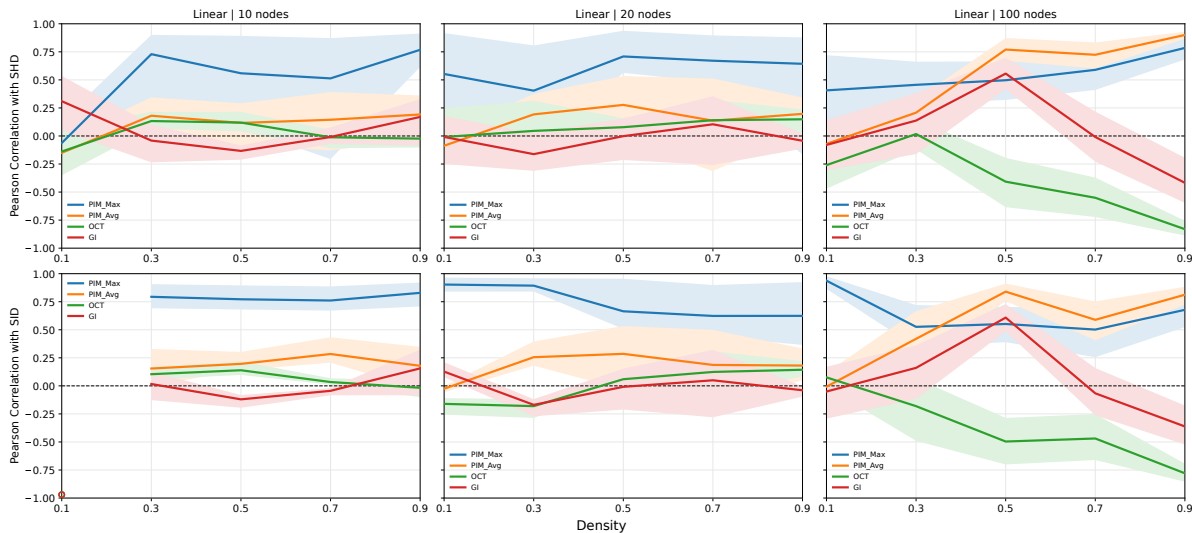

Figure 5: Correlation between each measure and SHD/SID in LiNGAM in different graph sizes and densities. Missing SID values for graphs of size 10 and with 0.1 density can be explained by near-perfect SHD scores, which result in a constant SID of 0 and undefined correlation.

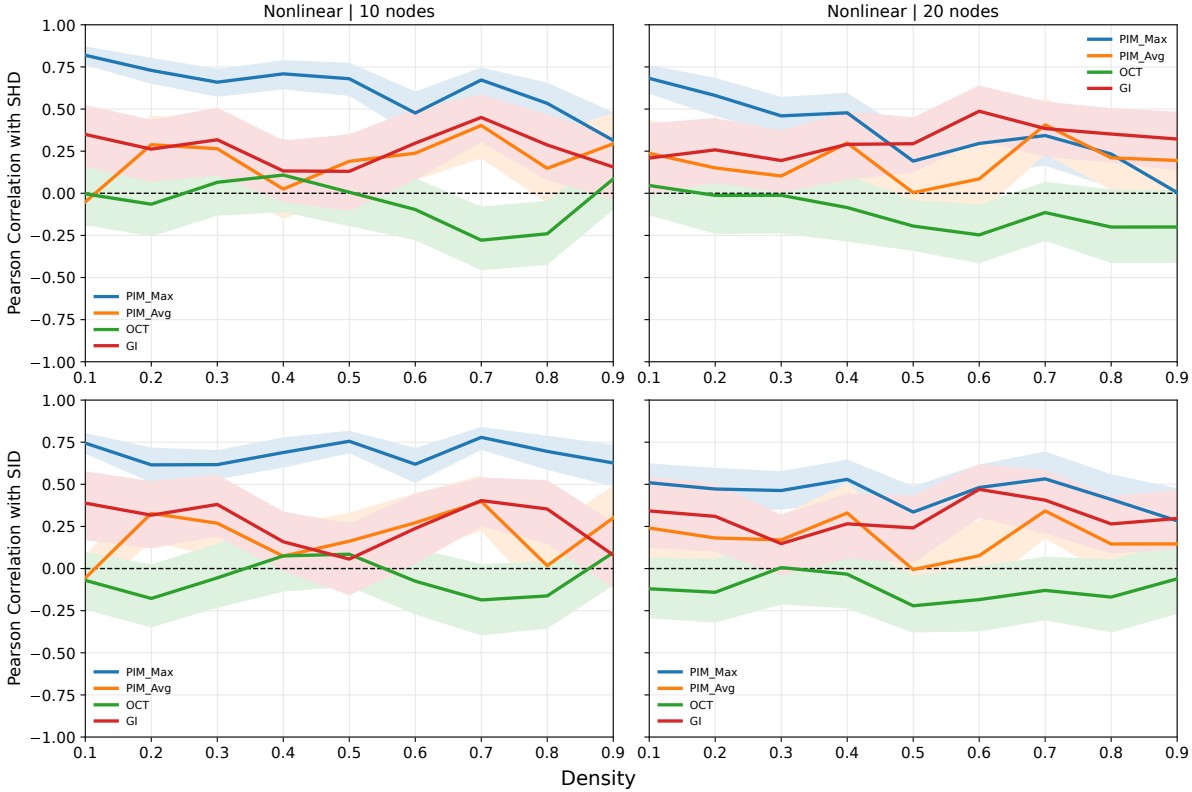

Figure 6: Correlation between each measure and SHD/SID in ANM.

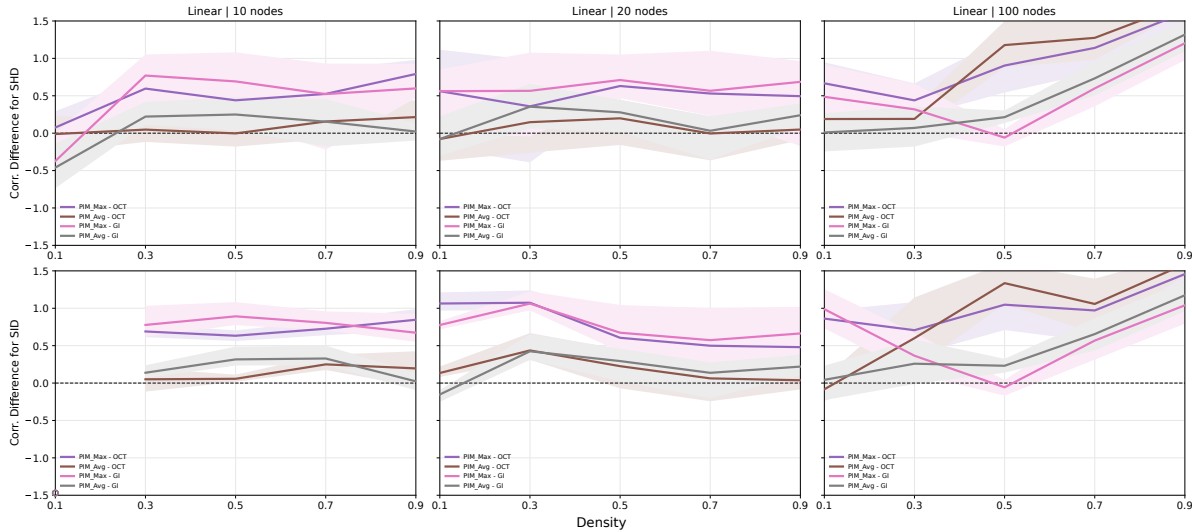

Figure 7: Difference of correlations between each measure and SHD/SID in LiNGAM, using bootstrapping. Here, the correlation between each score and SHD/SID is calculated, and the difference in correlations between our proposed scores and baselines is used to test whether our scores are more correlated with the reference. A confidence interval that excludes zero provides evidence of a significant difference.

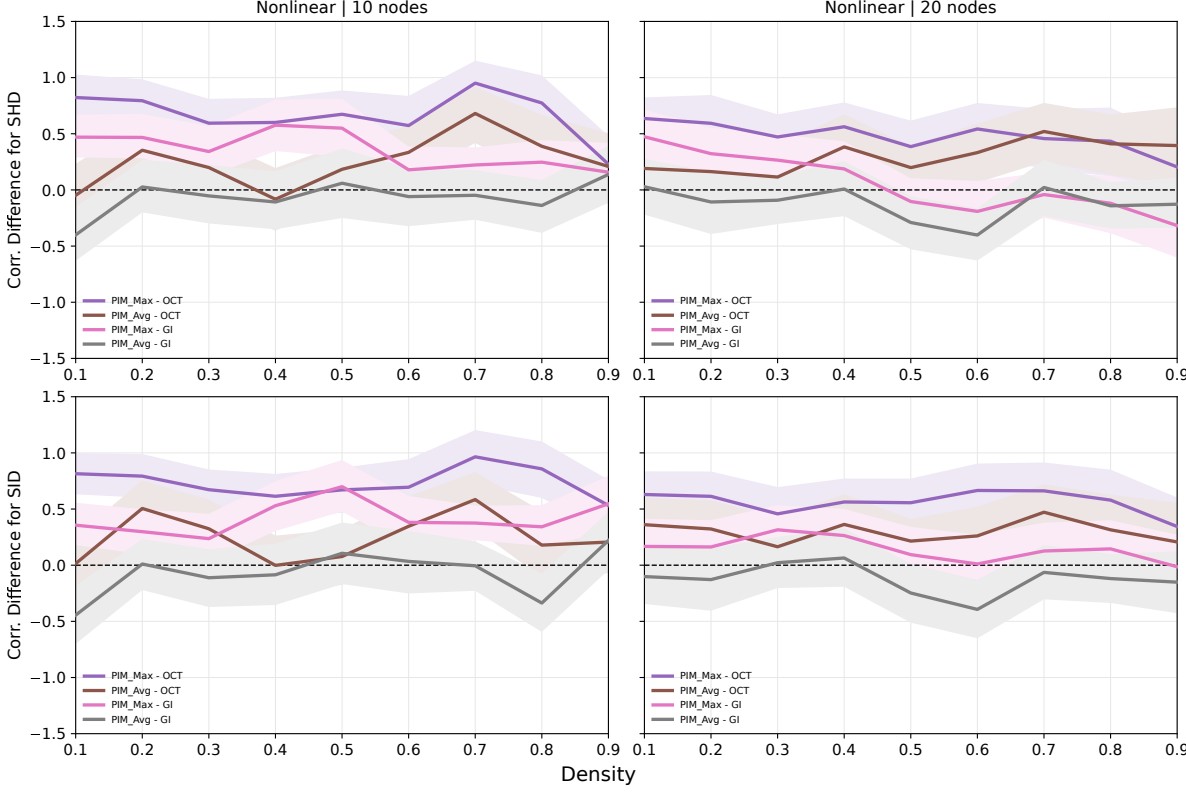

Figure 8: Difference of correlations between each measure and SHD/SID in ANM, using bootstrapping. Here, the correlation between each score and SHD/SID is calculated and the difference between correlations of our proposed scores and baselines is used to test whether our score is more correlated to the reference or not. A confidence interval that excludes zero provides evidence of a significant difference.

|  | SHD | SID | GI | $\text{PIM}_{\text{Max}}$ | $\text{PIM}_{\text{Avg}}$ | OCT |
|---|---|---|---|---|---|---|
| DirectLiNGAM | 36 | 63 | 4.1 | -0.019 | -3.662 | -0.808 |
| ICA-LiNGAM | 41 | 57 | 6.275 | 0.308 | -3.689 | -0.803 |
| NOTEARS | 26 | 69 | 2.625 | 0.308 | -3.164 | -0.757 |
| GOLEM | 26 | 75 | 4.95 | -0.159 | -3.491 | -0.644 |
| Ground Truth | 0 | 0 | N/A | -0.293 | -2.561 | -0.594 |

Table 1: Each algorithm and value for each evaluation measure. (Lower is better for all measures.)

The first row in Figure 5, and Figure 6 shows the correlation between each of the evaluation scores and SHD, and the second row shows the correlation with SID. In the case where there is missing data for SID (Figure 5, 10 nodes with 10% density), the causal discovery algorithm found all graphs near perfect; therefore, SHD for all but a few is 0 and SID is constantly 0; therefore, the correlation with SID is undefined. In the linear setting (Figure 5), it can be observed that $\text{PIM}_{\text{Max}}$ maintains a high correlation with both SHD and SID, except for graphs with 10 nodes with 10% density, where the difference in algorithm and theorem (not checking for causal minimality) has the most effect. GI also has a rise in performance for graph size 100 and density of 50%, which then drops sharply. $\text{PIM}_{\text{Avg}}$ shows a rise in performance in graph size 100 and higher densities. This higher correlation in graph size of 100 suggests that $\text{PIM}_{\text{Avg}}$ may be better suited to larger and denser graphs. It must be noted that both $\text{PIM}_{\text{score}}$ measures show stronger alignment with SHD and SID in more challenging scenarios with denser and larger graphs.

Figure 7 confirms that $\text{PIM}_{\text{Max}}$ with the mentioned exception always performs better (positive difference in correlation and zero outside confidence bounds) or similar to the baselines (positive difference in correlation and zero inside confidence bounds).

In a nonlinear setting (Figure 6), it can be seen that $\text{PIM}_{\text{Max}}$ maintains a higher correlation in 10 node setting, but in 20 node setting, a drop in performance in denser graphs is observable, which can be a result of the regression method choice. While OCT has a correlation of approximately zero constantly, $\text{PIM}_{\text{Avg}}$ and GI maintain a low positive correlation.

Figures 7 and 8 visualize the differences between the correlation of measures with SHD/SID and respective confidence bounds. We specifically subtracted the correlation of our measures ($\text{PIM}_{\text{Max}}$ and $\text{PIM}_{\text{Avg}}$) from that of the baselines. Positive differences point to the strength of our measures. It can be seen that both $\text{PIM}_{\text{score}}$ have a positive difference with OCT in all cases (except for graphs with 10 nodes 10% density) in the LiNGAM setting. $\text{PIM}_{\text{Max}}$ outperforms GI in most scenarios, with stronger evidence of correlation with SID, a more important quality for downstream causal inference tasks, specifically its ability to predict intervention outcomes. $\text{PIM}_{\text{Max}}$ is significantly more correlated with SHD and SID than OCT and GI in most settings, and $\text{PIM}_{\text{Avg}}$ in some cases is more correlated.

## 5.3 Real-world data

As a more practical evaluation, we explored various causal discovery algorithms, DirectLiNGAM (Shimizu et al., 2011), ICA-LiNGAM (Shimizu et al., 2006), NOTEARS (Zheng et al., 2018), and GOLEM (Ng et al., 2020) on the Sachs dataset (Sachs et al., 2005), which has a ground truth available, and evaluated the quality of the causal graph which is output of each algorithm using the proposed measures and baselines. As in real-world scenarios, we only observe the score; therefore, the best graph is selected according to the definition of the best score. All scores are adjusted to be lower-better, and as both SHD and SID are distances, a higher positive correlation is desired.

Table 1 shows discovered graphs based on different measures. As can be seen, each measure has a different range and cannot necessarily be interpreted like SHD. (It should be noted that these algorithms were not tuned, and their default values were used). Table 2 shows the Spearman correlation of each evaluation method with SHD and SID. $\text{PIM}_{\text{Max}}$ shows the highest correlation in both SHD and SID.

| Spearman Correlation | GI | $\text{PIM}_{\text{Max}}$ | $\text{PIM}_{\text{Avg}}$ | OCT |
|---|---|---|---|---|
| SHD | 0.63 | 0.71 | -0.97 | -0.87 |
| SID | -0.40 | 0.21 | -0.10 | -0.10 |

Table 2: Spearman correlation of evaluating each causal discovery algorithm with SHD and SID.

## 6 Conclusion

In this paper, we posed two problems: (i) causal graph verification ($G_{\text{ver}}$) and (ii) causal graph ranking ($G_{\text{rank}}$). Our primary goal was to address these problems without access to ground truth, making them applicable to causal discovery evaluation in real-world application scenarios. We proposed $\text{PIM}_{\text{score}}$, based on the Principle of Independent Mechanisms, for LiNGAMs and ANMs. To address the above-mentioned problems, we prove that, in the identifiable LiNGAM and restricted ANM settings considered here, the true graph is the unique candidate with jointly independent population residuals, attaining the lowest $\text{PIM}_{\text{score}}$ (Theorem 1). We proposed an algorithm to utilize our theoretical findings in empirical settings. In this algorithm, we regress each variable on its parents and calculate the residuals. Then, using pairwise mutual information, we look for dependencies. While we use pairwise mutual information and the presence of one dependency could answer the $G_{\text{ver}}$ problem, the whole mutual information matrix (Algorithm 1, line 7) represents an overview of residual dependencies in the whole graph and can be used to address the $G_{\text{rank}}$ problem using an aggregation function. Our experiments cover a wide range of settings, including graphs of different sizes and densities. Our results showed that the proposed scores, $\text{PIM}_{\text{Max}}$ and $\text{PIM}_{\text{Avg}}$, can be applied to both problems. In particular, $\text{PIM}_{\text{Max}}$ shows stronger correlation with SHD and SID than the considered baselines in many tested settings. Based on these results, we conclude that our proposed measure $\text{PIM}_{\text{Max}}$ is a promising proxy for both SHD and SID, making it a promising measure for evaluating recovered graphs when the true graph is unavailable.

One limitation of our work is the lack of support for latent confounding. In addition, our experiments point to several promising directions for future research. First, the algorithm and the proposed scores could be extended to incorporate regression weights and detect non-minimal graphs more explicitly. Second, based on the results, it would be valuable to investigate alternative aggregation functions such as high quantiles and Top-K average. Finally, an important practical direction is to study how these measures perform in algorithm configuration settings, such as hyperparameter tuning and model selection.

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

## A   Appendix

**Lemma A.1.** *Referring to LiNGAM model, Equation 3, we have:*

$$\mathbf{X} = (\mathbf{I} - \boldsymbol{B})^{-1}\mathbf{N} \tag{8}$$

*where $X$ is the vector of all variables, $\mathbf{I} - \boldsymbol{\beta}$ is of full rank, and selection of $k$ distinct rows from it results in a rank $k$ matrix. Therefore, $\boldsymbol{XX}^T$ is of full rank and has no zero eigenvalues. Knowing covariance matrices are positive semi-definite, with no zero eigenvalues, $E[\boldsymbol{XX}^T]$ is positive definite and invertible.*

**Proposition A.2.** *Let $pa_G(X_i)$ be the parent set of variable $X_i$ in the given graph $G$. If $pa_G(X_i)$ is the correct underlying parent set, the residual of the regression of $X_i$ on $pa_G(X_i)$ is independent of $pa_G(X_i)$.*

*Proof.* Consider any variable $Y$:

$$Y = \sum_{X_j \in Pa_G(Y)} \beta_j X_j + N_Y$$

We can write the above equation in the vector form where $X$ and $B^*$ contain $\{X_i\} \in Pa_G(Y)$ and $\{\beta_j^*\}$ $j \in Pa_G(y)$, respectively.

$$Y = X_{pa_G(X_i)}^T \boldsymbol{\beta} + N_Y$$

Where $\beta_j^*$ are the correct direct causal effect of $X_j$ on $Y$. Note that $Pa_G(Y) = Pa_{G^*}(Y)$ due to the assumption in the statement of the proposition.

Let $\boldsymbol{\beta}$ be the coefficient of linear regression of $Y$ on $\mathbf{X}$ using Lemma A.1 we have:

$$E[\boldsymbol{\beta}] = E[\mathbf{XX}^T]^{-1}E[\mathbf{X}Y]$$

$$\begin{aligned}
E[\boldsymbol{\beta} - \boldsymbol{\beta}^*] &= E[\mathbf{XX}^T]^{-1}E[\mathbf{X}Y] - \boldsymbol{\beta} \\
&= E[\mathbf{XX}^T]^{-1}E[\mathbf{X}(\mathbf{X}^T\boldsymbol{\beta}^* + N_Y)] - \boldsymbol{\beta}^* \\
&= \boldsymbol{\beta}^* + E[\mathbf{XX}^T]^{-1}E[\mathbf{X}N_Y] - \boldsymbol{\beta}^* \\
&= E[\mathbf{XX}^T]^{-1}E[\mathbf{X}N_Y]
\end{aligned} \tag{9}$$

$$E[\boldsymbol{\beta} - \boldsymbol{\beta}^*|\mathbf{X}] = E[E[(\mathbf{XX}^T)]^{-1}E[\mathbf{X}N_y]|\mathbf{X}] \tag{10}$$
$$= (\mathbf{XX}^T)^{-1}\mathbf{X}E[N_y|\mathbf{X}]$$
$$= (\mathbf{XX}^T)^{-1}\mathbf{X}E[N_y] = 0$$

Therefore, from Equations 9 and 10 we can imply that $\boldsymbol{\beta} = \boldsymbol{\beta}^*$. Furthermore,

$$Y - \hat{Y} = \mathbf{X}^T\boldsymbol{\beta}^* + N_Y - \mathbf{X}^T\boldsymbol{\beta} = N_Y \tag{11}$$

By the principle of independent mechanisms, we know the residuals are independent of the regressand. Hence $N_Y \perp\!\!\!\perp X$ and the proof is complete. $\qquad\square$

**Proposition A.3.** *For a given graph $G$ and a variable $Y$, if $Pa_{G^*}(Y) \subset Pa_G(Y)$, then the true parent set of $Y$ can be recovered from regressing $Y$ on $Pa_G(Y)$.*

*Proof.* Consider the square loss in linear regression:

$$S(\boldsymbol{\beta}) = E[(Y - X^T\boldsymbol{\beta})^2] \tag{12}$$

Any critical point of $S(\boldsymbol{\beta})$ should satisfy the following condition:

$$\frac{\partial S(\boldsymbol{\beta})}{\partial \boldsymbol{\beta}} = 2E[(Y - X^T\boldsymbol{\beta})\mathbf{X}] = 0 \tag{13}$$

For the vector with correct values for $Pa_{G^*}(Y)$ and zero entries for other coordinates, the above condition (Equation 13) is satisfied. Moreover by taking the second derivative of the error term, we have:

$$\frac{\partial^2 S(\boldsymbol{\beta})}{\partial \hat{\beta}^2} = 2E[\mathbf{XX}^T] \tag{14}$$

By Lemma A.1 we know that $XX^T$ is positive definite, and based on the Second-Order Sufficient Condition (SOSC) for optimality, the vector described above is the unique minimizer of Equation 12 and nonzero entries correspond to $Pa_{G^*}(Y)$. $\qquad\square$

**Theorem A.4.** *Residuals of regression of all variables on their parents in a graph $G$ are independent of the parents and there are no zero coefficients in the regression, if and only if the given graph is the true underlying graph $G$.*

*Proof.* We write this proof for the parent set, by changing to ancestor, everything still holds and some cases simplify. There is one case we cannot prove for parent set but it holds for the ancestors.

By Proposition A.2, we know the true graph has this property, in case the given graph is not the true graph; Then there exists at least one variable that either has extra parents, missing parents, or both. Without loss of generality, we assume that the variable is $Y$. We denote the set of extra variables $EPa_G(Y)$ as the set of variables that are in $Pa_G(Y)$ but not in $Pa_{G^*}(Y)$, and set of missing variables $MPa_G(Y)$ as the set of variables that are in $Pa_{G^*}(Y)$ but not in $Pa_G(Y)$.

$$Y = \sum_{X_i \in Pa_{G^*}(Y)} \beta_i^* X_i + N_Y \tag{15}$$

We divide this into following cases:

1. Extra parent, without any missing parent

2. Non-ancestor extra parent

3. Non-ancestor missing parent

4. Ancestor missing parent, without extra parents

5. Ancestor extra parent, with missing parents

**1- Extra parent, without any missing parent**

In this case, Proposition A.3 applies, and the extra parents can be detected by the zero weights.

**2- Non-ancestor extra parent**

In the case of non-ancestor extra parent, in a graph $G$, consider a variable $Y$ that has at least one extra parent $X_j$ that is non-ancestor of $Y$ in the parent set ($X_j \notin An_{G^*}(Y)$. From the set of non-ancestor extra parents, we pick $X_j$ such that it is not an ancestor of any other variable in that set, i.e. $X_j \in \{EPa_G(Y) \setminus An_{G^*}(EPa_G(Y))\}$ By regressing $Y$ on $Pa_G(Y)$ we have:

$$
\begin{aligned}
Y - \hat{Y} &= \sum_{X_i \in Pa_{G^*}(Y)} \beta_i^* X_i + N_Y - \sum_{X_i \in Pa_G(Y)} \beta_i X_i \\
&= \sum_{X_i \in Pa_G(Y) \setminus EPa_G(Y)} (\beta_i^* - \beta_i) X_i + N_Y \\
&+ \sum_{X_i \in MPa_G(Y)} \beta_i^* X_i - \sum_{X_i \in \{EPa_G(Y) \setminus X_j\}} \beta_i X_i - \beta_j X_j
\end{aligned}
\tag{16}
$$

$X_j$ can be written as a linear combination of exogenous noises corresponding to its ancestors:

$$
X_j = \sum_{p \in an_{G^*}(X_j)} \alpha_p N_p + N_j
\tag{17}
$$

Where $\alpha_p$'s are some real value constants.

As $X_j$ is non-ancestor for $Y$ and also is not an ancestor of any other variable in the extra variables, the term $N_j$ only exists in the last term of Equation 16. Therefore, we have:

$$
Y - \hat{Y} = \cdots - \beta_j X_j = \cdots - \beta_j N_j
\tag{18}
$$

Based on Darmois-Skitovic Theorem (Darmois, 1953) we have $\ldots + \beta_j N_j \not\perp\!\!\!\perp X_j$

**3- Non-ancestor missing parent**

In case of non-ancestor missing parent, in a graph $G$, consider a missing parent $X_j$, $X_j \in Pa_{G^*}(Y)$ and $X_j \notin Pa_G(Y)$, that has no descendant in the missing parents, i.e. $De_{G^*}(X_j) \cap MPa(Y) = \varnothing$, is a missing parent for $Y$.

By regressing $Y$ on its parents in $Pa_G(Y)$, we have:

$$
\begin{aligned}
Y - \hat{Y} &= \sum_{X_i \in Pa_{G^*}(Y)} \beta_i^* X_i + N_Y - \sum_{X_i \in Pa_G(Y)} \beta_i X_i \\
&= \sum_{X_i \in Pa_G(Y)} (\beta_i^* - \beta_i) X_i + N_Y + \sum_{X_i \in MPa_G(Y)} \beta_i^* X_i
\end{aligned}
\tag{19}
$$

By checking the dependency of the residual and $X_j$, we have:

$$
\begin{aligned}
Y - \hat{Y} &= \sum_{X_i \in Pa_G(Y)} (\beta_i^* - \beta_i) X_i + N_Y + \sum_{X_i \in MPa_G(Y)} \beta_i^* X_i \\
&= \sum_{X_i \in Pa_G(Y)} (\beta_i^* - \beta_i) X_i + N_Y + \\
&\quad \sum_{X_i \in \{MPa_G(Y) \setminus \{X_j\}\}} \beta_i^* X_i + \beta_j^* X_j
\end{aligned}
\tag{20}
$$

Based on Darmois-Skitovic Theorem (Darmois, 1953) we have: $\ldots + \beta_j^* N_j \not\perp\!\!\!\perp \cdots + N_j$

### 4- Ancestor missing parent

In case of an ancestor missing parent, in a graph $G$, consider a variable $Y$ that has missing parents in the parent set and no extra parents. Suppose that $X_j$ is the ancestor missing parent. $X_j \in MPa_G(Y)$ and $|De_{G^*}(X_j) \cap An_{G^*}(Y)| > 0$ assume a causal order over $\{De_{G^*}(X_j) \cap An_{G^*}(Y)\}$. We take the last variable in the causal order, $X_m$:

As $N_j$ is present in another parent $X_m$ ($X_j$ has more than one directed path to $Y$), we have:

$$
\begin{aligned}
Y - \hat{Y} &= \sum_{i \in \{Pa_G(Y) \backslash m\}} (\beta_i^* - \beta_i) X_i + \sum_{i \in \{MPa_G(Y) \backslash j\}} \beta_i^* X_i \\
&+ (\beta_m^* - \beta_m) \left( \sum_{p \in \mathrm{An}_G(X_m)} \alpha_p N_p + N_m \right) \\
&+ \beta_j \left( \sum_{p \in \mathrm{an}_G(X_j)} \alpha_p N_p + N_j \right) + N_Y \\
&= \cdots + (\beta_m^* - \beta_m) N_m + (\cdots + (\beta_m^* - \beta_m)\alpha_m + \beta_j^*) N_j
\end{aligned}
\tag{21}
$$

If $\beta_m$ has any value but $\beta_m^*$, then $N_m$ will remain in the residual, and a dependency will arise between the residual and $X_m$. If $\beta_m$ is equal to $\beta_m^*$, then the effect of $X_j$ on $Y$ through $X_m$ is canceled. Now we remove $X_m$ from the causal order and repeat this procedure until only one variable remains in the order. If all the $\beta_m^*; m \in \{De_{G^*}(X_j) \cap An_{G^*}(Y)\}$ except one is recovered, then we have one last variable $X_m$:

$$
\begin{aligned}
Y - \hat{Y} &= \sum_{i \in \{Pa_G(Y) \backslash m\}} (\beta_i^* - \beta_i) X_i + \sum_{i \in \{MPa_G(Y) \backslash j\}} \beta_i^* X_i \\
&+ (\beta_m^* - \beta_m) \left( \sum_{p \in \mathrm{an}_G(X_m)} \gamma_p N_p + N_m \right) \\
&+ \beta_j \left( \sum_{p \in \mathrm{an}_G(X_j)} \alpha_p N_p + N_j \right) + N_Y \\
&= \cdots + (\beta_m^* - \beta_m) N_m + ((\beta_m^* - \beta_m)\alpha_m + \beta_j^*) N_j
\end{aligned}
\tag{22}
$$

If $\beta_m$ is not equal to $\beta_m^*$ then a dependency will arise between the residual and $X_m$, and if it is equal to $\beta_m^*$, then $N_j$ will remain in the residual, and a dependency will arise between $X_j$ and the residual.

$$
\begin{cases}
\beta_m^* - \beta_m = 0 \\
(\beta_m^* - \beta_m)\gamma + \beta_j = 0
\end{cases}
\implies
\begin{aligned}
\beta_m &= \beta_m^* \\
\beta_m &= \beta_m^* + \frac{\beta_j}{\gamma}
\end{aligned}
\tag{23}
$$

$\beta_m$ cannot take a value that makes both coefficients zero, therefore at least one out of $N_m$ or $N_j$ remains in the residual and will have a dependency with $X_m$ or $X_j$.

### 5- Ancestor extra parent with missing parent

Consider $X_j, X_j \in EPa_G(Y)$, if $De_{G^*}(X_j) \cap MPa_G(Y) = \{\varnothing\}$, then effect of $X_j$ will be canceled via the descendants in regression and it can be detected by a zero coefficient.

In case $De_{G^*}(X_j) \cap MPa_G(Y) \neq \{\varnothing\}$, if there exists one variable $X_m \in De_{G^*}(X_j) \cap MPa_G(Y)$ that satisfy the following condition, $X_m \notin An_{G^*}(Pa_{G^*}(Y))$, then we have:

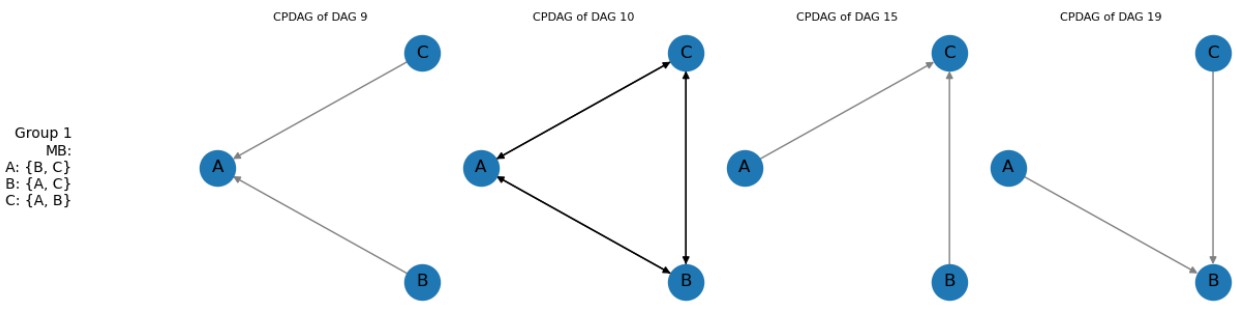

Figure B.1: An example of CPDAGs that share the same Markov Boundary and get the same score from OCT in graph sizes of three.

$$
\begin{aligned}
Y - \hat{Y} = & \sum_{i \in \{Pa_G(Y) \setminus EPa_G(Y)\}} (\beta_i^* - \beta_i) X_i + \\
& \sum_{i \in \{MPa_G(Y) \setminus X_m\}} \beta_i^* X_i - \sum_{i \in \{EPa_G(Y)\} \setminus X_j} \beta_i X_i \\
& + \beta_m \left( \sum_{p \in \mathrm{an}_G(X_m)} \gamma_p N_p + N_m \right) \\
& - \beta_j \left( \sum_{p \in \mathrm{an}_G(X_j)} \alpha_p N_p + N_j \right) + N_Y \\
= & \cdots + (\beta_m) N_m \not\!\perp Y
\end{aligned}
\tag{24}
$$

We know that $X_m$ does not have a descendant in the parent set of $Y$; therefore, the exogenous noise $N_m$ will raise a dependency.

By now, we have proved every case with regression on the parent set, and the only remaining case is the case that we have an ancestor extra parent with missing parents, where at least a descendant of the extra parent is a missing parent, and none of the variables that are descendants and missing parents have only one path to the variable.

For the last case, by changing the regression to the ancestor set, we will have the missing parent in the regression, and by looking at the weights, we will know that the graph is not the true graph.

By this, we conclude our proof. □

This proof also shows how different errors in graph creates dependency.

## B  OCT Similarly Scored Graphs

Figure B.1 shows an example of CPDAGs sharing the same Markov Boundary and getting the same score from OCT in graph sizes of three. In this class of graphs, they all get the same score as OCT relies on MB to compute predictive performance.

## C  Effect of sample size on methods

Figures C.1-C.4 show the performance of different measures on the same graphs with different data sizes. While all of them exhibit variation within the same ranges, an interesting point can be seen. $\mathbf{PIM}_{\mathrm{Avg}}$ shows

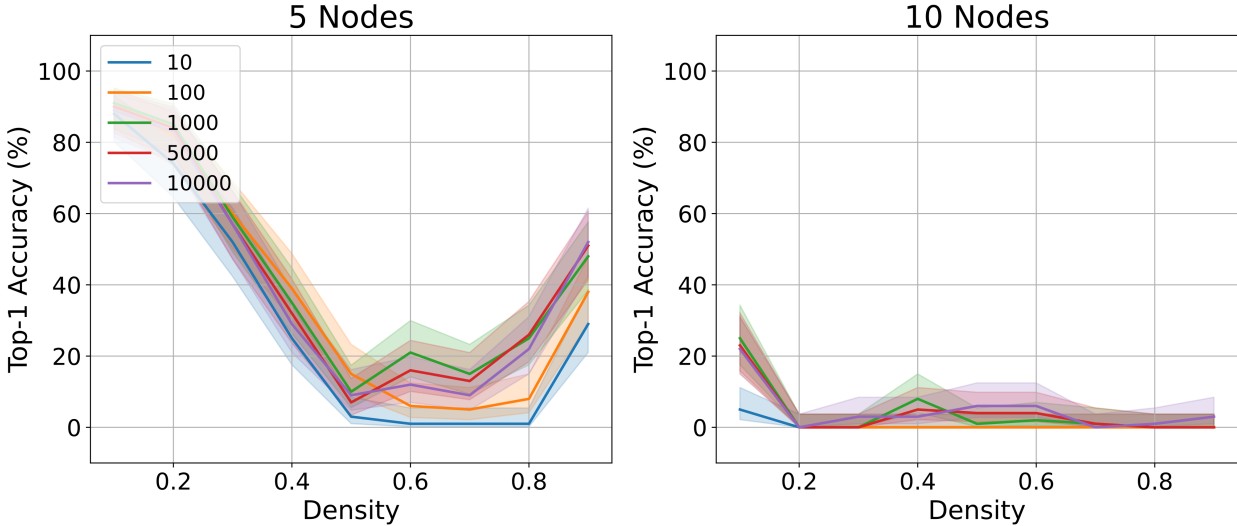

Figure C.1: Accuracy of identifying the ground truth among similar graphs in LiNGAM (linear setting) setting using OCT with different sample sizes, for graphs with size 5 (left) and 10 (right).

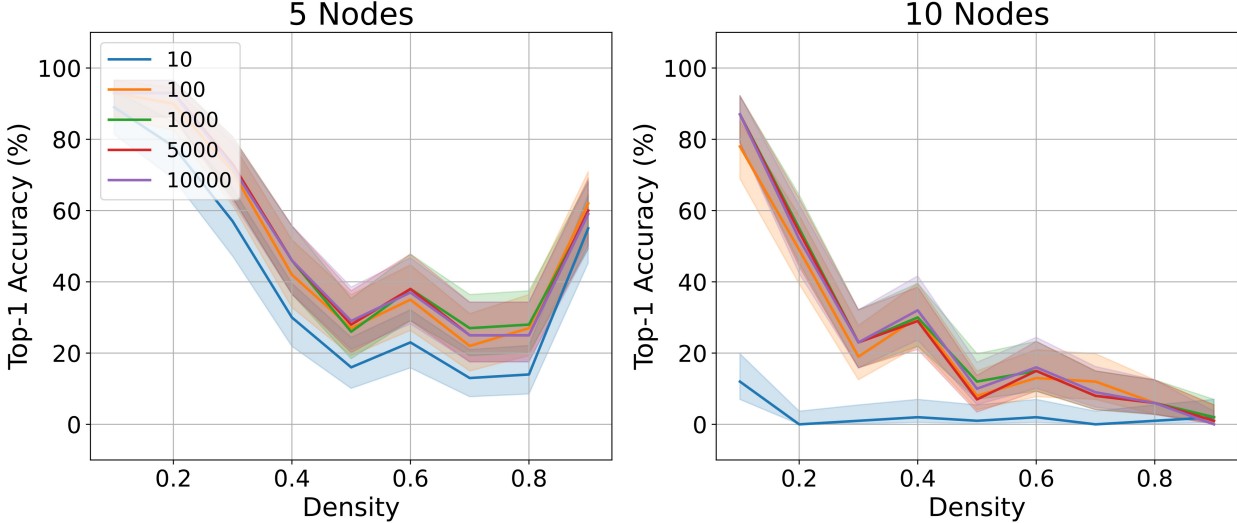

Figure C.2: Accuracy of identifying the ground truth among similar graphs in LiNGAM (linear setting) setting using GI with different sample sizes, for graphs with size 5 (left) and 10 (right).

no sign of a drop in performance, OCT shows a drop in denser graphs of 5 nodes with 10 and 100 samples, GI shows a drop in 10 sample case, and $\mathbf{PIM}_{\mathrm{Avg}}$ shows a drop in 10 and 100 sample sizes.

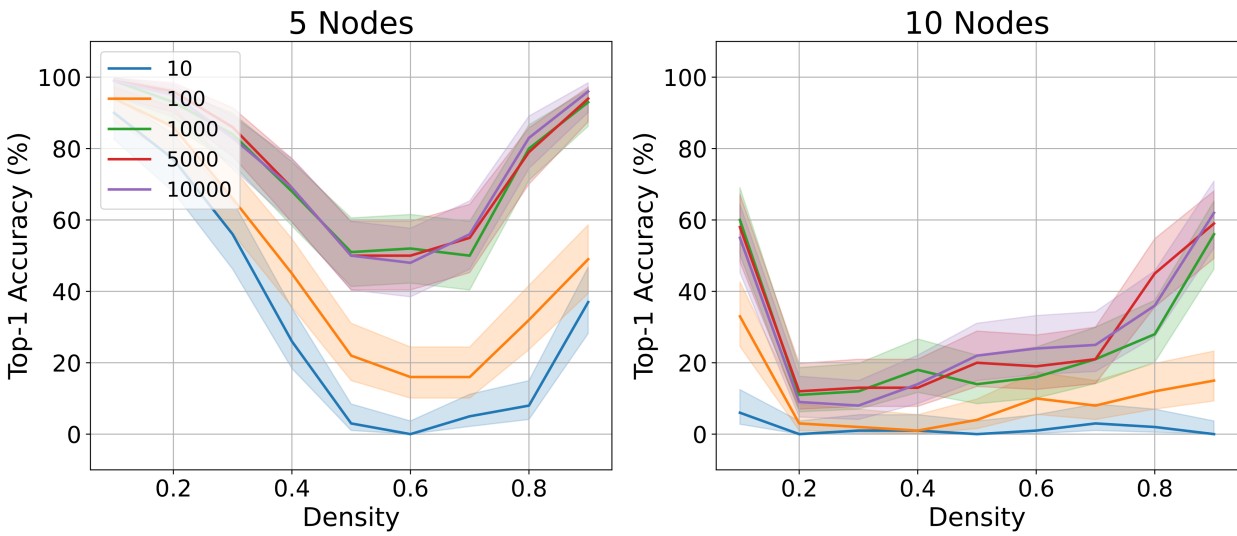

Figure C.3: Accuracy of identifying the ground truth among similar graphs in LiNGAM (linear setting) setting using $\text{PIM}_{\text{Max}}$ with different sample sizes, for graphs with size 5 (left) and 10 (right).

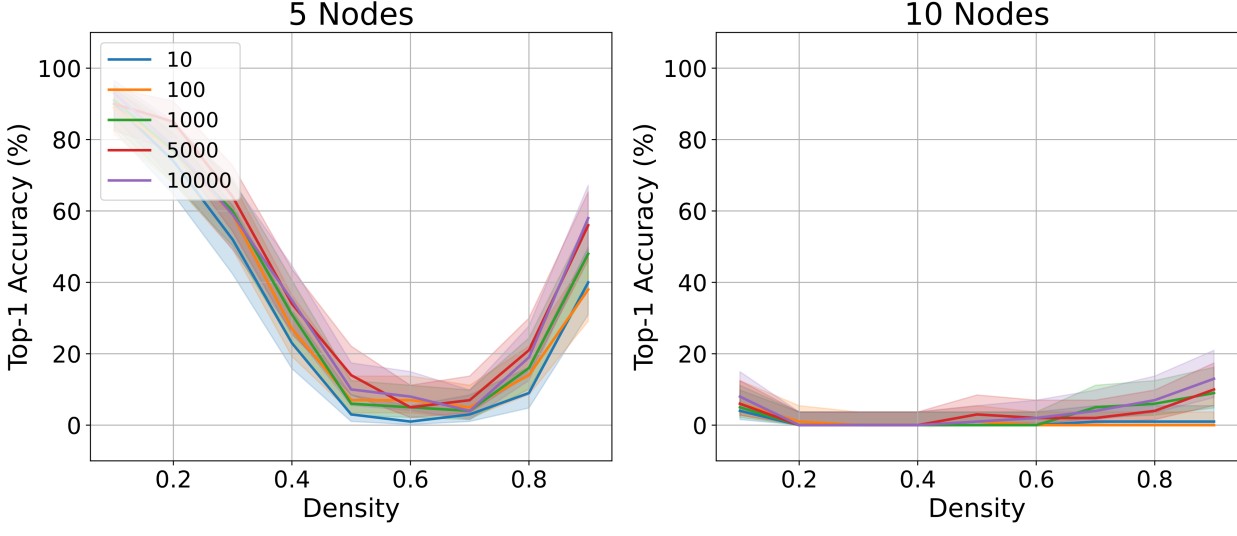

Figure C.4: Accuracy of identifying the ground truth among similar graphs in LiNGAM (linear setting) setting using $\text{PIM}_{\text{Avg}}$ with different sample sizes, for graphs with size 5 (left) and 10 (right).

