# OpenReview forum: "Evaluating Causal Discovery Algorithms Without Ground Truth in Additive Noise Models"
_TMLR — Under review for TMLR_

### Review · Reviewer_UYm1 · 2026-06-16

**Summary Of Contributions:**

**Summery**:

This paper proposes PIMscore, a ground-truth-free score for evaluating and ranking causal discovery outputs under LiNGAM and restricted additive noise model assumptions. The main idea is that, for the correct graph, the residuals obtained by regressing each variable on its parents should be mutually independent. The paper proves this property at the population level under its assumptions and implements the score using pairwise mutual information between residuals. Experiments on synthetic data and the Sachs dataset suggest that PIM_Max often correlates better with SHD and SID than OCT and graphical incompatibility.


**Strengths**:

* The paper studies an important practical problem: evaluating causal discovery results without ground truth, which is especially relevant for scientific domains such as biological or gene regulatory data.
* The proposed PIMscore is simple and intuitive, and the paper is generally easy to follow.
* The paper clearly defines graph verification and graph ranking, and evaluates the method in both linear and nonlinear settings.

**Weaknesses**:

* The theoretical guarantee is derived under quite strong assumptions, including LiNGAM or restricted ANM assumptions, causal sufficiency, causal minimality, and an additional assumption that the residual-induced representation for a candidate graph remains in an identifiable and causally minimal model class. These assumptions make the setting close to one where the true causal graph is already identifiable in principle.

* Related to the first point, the paper should better justify the need for an additional evaluation score under these assumptions. If the data are assumed to follow identifiable LiNGAM or restricted ANM models, it is not fully clear why one should use PIMscore instead of directly applying existing identifiable causal discovery methods or model-based graph scoring methods.

* If the goal is to justify PIMscore as a practical finite-sample evaluation tool, then the paper seems lack a finite-sample analysis. The score depends on estimated regressions, residuals, and mutual information, but the paper does not study how these estimation errors affect graph verification or ranking. It would be useful to provide either theoretical convergence results or empirical sample-size ablations. The current experiments mainly vary graph size and density, but do not sufficiently examine how PIMscore behaves as the sample size changes.

* More baseline scores should be included. OCT and graphical incompatibility are reasonable ground-truth-free heuristics, but they are not the most natural baselines under the paper's LiNGAM/ANM assumptions. Model-based scores such as AIC/BIC-style SEM scores, LiNGAM likelihood or ICA-style scores, ANM likelihood scores, and residual-independence measures such as HSIC/KCI or distance correlation would make the comparison more convincing.

**Audience:**

Yes

**Audience Explanation:**

Causal discovery is an important topic in the machine learning community, and evaluating learned causal graphs without ground truth is a practical and relevant problem.

**Broader Impact Concerns:**

I think no broader impact concerns should be specifically discussed here.

**Claims And Evidence:**

No

**Claims Explanation:**

The paper presents a reasonable and interesting idea, but I think the current evidence only partially supports the main claims. The experiments are mostly conducted on synthetic LiNGAM/ANM data with a large fixed sample size, while there is no sample-size ablation to show how PIMscore behaves in realistic finite-sample settings. The comparisons are also mainly limited to OCT and GI, without including more natural model-based or residual-independence baselines. Therefore, in my opinion, the paper would benefit from a clearer theoretical and empirical justification of when PIMscore is needed and why it should be preferred over existing alternatives.

**Requested Changes:**

I would suggest that the authors clarify more explicitly why PIMscore is needed under LiNGAM/ANM assumptions, and discuss how PIMscore differs from or improves over existing model-based scores in this setting. I would also encourage the authors to include more relevant baselines, such as AIC/BIC-style SEM scores, LiNGAM/ICA-style scores, ANM likelihood scores, HSIC, or distance correlation, and to add sample-size ablation experiments to better show the finite-sample behavior of PIMscore.

---

### Review · Reviewer_xXhC · 2026-06-17

**Summary Of Contributions:**

As suggested in the title, the largest contribution of this paper is to propose a method to evaluate the performance of causal discovery using only observational data, i.e., without the need of groundtruth. Theoretically, it proves that under certain conditions, only the residuals are independent if and only if the candidate graph is the true causal graph. Empircially, the authors show the proposed PIMMax consistently outperform baselines on medium and denser graphs, and it shows stronger correlation with SHD and SID.

**Audience:**

Yes

**Audience Explanation:**

The paper addresses a practically important question: for a real-world dataset, different causal discovery methods often produce different causal graphs. However, how can we determine which graph is more credible, or at least more reasonable, when the ground-truth causal structure is unknown?

Most research in the causal discovery community focuses on theoretical questions, such as identifying the assumptions under which certain levels of causal recovery are achievable. In contrast, the evaluation problem has received much less attention, despite being highly relevant in practice. In nearly all real-world applications, the true causal graph is unavailable, making it difficult to assess the quality of discovered structures. Therefore, the problem targeted by this paper is itself meaningful and likely to attract broad interest from the community.

Although the proposed algorithm is relatively simple and somewhat heuristic, which may explain why the PIM score does not perform particularly well on sparse graphs, the importance of the problem being addressed is sufficient to justify publication.

**Broader Impact Concerns:**

There are no concerns on the ethical implications of this work.

**Claims And Evidence:**

Yes

**Claims Explanation:**

The main theoretical claims of this paper "The residuals are independent if and only if the candidate graph is the true causal graph" is intuitional, and the authors also show the algorithm motivated by the independence mechanism shows superior performance using synthetic dataset and real-world dataset.

**Requested Changes:**

The performance of PIMscore in the nonlinear setting appears to depend strongly on the choice of the regression model used to estimate residuals. The paper attributes some of the performance degradation to the use of a shallow ReLU network, but this claim is not systematically investigated. I encourage the authors to conduct a sensitivity analysis with respect to the regression architecture (e.g., varying activation functions, and other universal function approximators) to assess the robustness of the proposed method. Such an analysis would help clarify whether the observed results reflect limitations of PIMscore itself or simply the quality of the regression model used to estimate the residuals.

---

### Review · Reviewer_UPGr · 2026-07-11

**Summary Of Contributions:**

The paper proposes PIMscore, a practical unsupervised evaluation score for evaluating and ranking candidate causal graphs when the ground-truth graph is unavailable. The key leverage is the Principle of Independent Mechanisms. For each candidate graph, it regresses every variable on its proposed parents and computes the residuals. Under LiNGAM or ANM assumptions, the correct graph should recover the independent exogenous noise terms as residuals, while incorrect graphs should leave dependence among residuals. The method therefore measures pairwise mutual information among residuals and uses the resulting dependence score to verify the true graph among candidates or rank graphs according to expected closeness to the unknown truth.

**Audience:**

Yes

**Audience Explanation:**

Evaluation of causal discovery performance, especially in an unsupervised setting, is an important problem and has been relatively under-studied. While the applicability of the current results is limited, the paper can provide a motivating starting point for the community to explore more seriously.

**Claims And Evidence:**

No

**Claims Explanation:**

### **1. Claims in Theorem 1**
Theorem 1 aims to justify PIMscore, by showing  a candidate graph is correct if and only if the residuals obtained by regressing each variable on its proposed parents are jointly independent.

The forward direction is standard: if the candidate graph is the true graph, the residuals recover the independent exogenous noises. The converse direction, however, relies on Assumption 2, which I find problematic.

The issue is residual independence alone does not exclude non-minimal supergraphs of the true graph, so PIMscore itself cannot distinguish graphs with spurious edges. Assumption 2 excludes this case by requiring that any candidate representation with independent residuals must already be causally minimal. This means Assumption 2 rules out exactly the ambiguous graphs that PIMscore itself cannot detect. This potentially explains for the poor empirical performance of PIMscore in low density graphs (Figures 1-4)

Technically, PIMscore is only theoretically valid within a restricted class of candidate graphs whose induced representations are causally minimal. In pratice, Assumption 2 is a relatively strong assumption and may not be guaranteed by LiNGAM, ANM, or observational data. So the contribution of PIMscore might have been overstated, if  not intepreted carefully.

### **2. Claims about Pairwise dependence**

A second caveat is that Theorem 1 is stated in terms of joint residual independence, whereas the implemented PIMscore measures only pairwise dependence. These are not equivalent conditions in general. Pairwise independence can miss higher-order dependence among three or more residuals. Thus, even if all pairwise mutual informations vanish, the residual vector need not be jointly independent. This means that the practical scores do not fully implement the independence condition required by the theorem.

### **3. Claims about PIMscore as proxy for SID and SHD**

I notice there are some empirical inconsistencies that can weaken the claim and ambiguate the reliability of the correlation results. In table 1, Sachs real example, PIM max favors GOLEM, which has worst SID and PIM avg assigns the ground-truth graph a worse score than all discovered graphs. However, there is often trade-off between SHD and SID themselves where SHD favors sparse graphs and SID favors dense graph. It would be more convincing if the authors could show that using PIMscore is more reliable and less ambiguous for evaluating causal discovery performance.

**Requested Changes:**

See above comments.

Additionally, what remains unclear to me is how the mutual information terms are computed from finite samples. Since the population residual densities are unknown, the empirical PIMscore must depend on some particular estimators, which potentially require additional parameteric  assumptions . This detail is important for reproducibility and for understanding the reliability of the score.